# Design, Synthesis, Molecular Docking, and Tumor Resistance Reversal Activity Evaluation of Matrine Derivative with Thiophene Structure

**DOI:** 10.3390/molecules26020417

**Published:** 2021-01-14

**Authors:** Jinrui Wei, Yuehui Liang, Lichuan Wu

**Affiliations:** 1Guangxi Scientific Research Center of Traditional Chinese Medicine, Guangxi University of Chinese Medicine, Nanning 530200, Guangxi, China; weijinrui5566@163.com; 2School of Chemistry and Chemical Engineering, Guangxi University, Nanning 530004, Guangxi, China; 1814404027@st.gxu.edu.cn; 3Medical College of Guangxi University, Nanning 530004, Guangxi, China

**Keywords:** nasopharyngeal carcinoma, drug resistance, matrine derivative, molecular docking

## Abstract

Nasopharyngeal carcinoma (NPC) frequently occurs in Southern China. The main treatments of NPC are chemotherapy and radiotherapy. However, chemo-resistance arises as a big obstacle in treating NPC. Therefore, there is a great need to develop new compounds that could reverse tumor drug resistance. In this study, eight matrine derivatives containing thiophene group were designed and synthesized. Structures of these 8 compounds were characterized by ^1^H-NMR, ^13^C-NMR, and high-resolution mass spectrometer (HRMS). The cytotoxicity and preliminary synergistic effects of these 8 compounds were detected against nasopharyngeal carcinoma (NPC) cells and cisplatin-resistant NPC cells (CNE2/CDDP), respectively. Furthermore, the in vivo and in vitro tumor resistance reversal effects of compound **3f** were evaluated. Moreover, docking studies were performed in Bclw (2Y6W). The results displayed that compound **3f** showed synergistic inhibitory effects with cisplatin against CNE2/CDDP cells proliferation via apoptosis induction. Docking results revealed that compound **3f** may exert its effects via inhibiting anti-apoptosis protein Bcl-w.

## 1. Introduction

Nasopharyngeal carcinoma (NPC) is a common head and neck cancer characterized with remarkable ethnic and geographic distributions which frequently occurs in Southern China, Southeast Asia, and Northern Africa while rarely in Western countries [1]. Surgery is one of the effective ways in the treatment of cancer. However, it is very difficult to apply surgery in the treatment of NPC due to its anatomic location [2]. Fortunately, NPC is a highly radio- and chemo-sensitive tumor, and NPC patients with early-stage and locally advanced are treated with radiotherapy single or combined treatment of chemo- and radiotherapy [3]. Although the combined chemo-radiotherapy produces a satisfying survival rate (85−90% for 5-year) [4,5], there are still 8−10% patients undergoing a recurrence and developing tumor metastasis [6,7]. For those recurrent NPC, the standard treatment is multi-drug chemotherapy with platinum [8]. However, platinum resistance arises a big obstacle in curing recurrent NPC patients [2]. Consequently, developing potential compounds that show synergistic effects with cisplatin (CDDP) are of great significance.

Matrine, which is the main component of Compound Kushen Injection, was approved to treat various kinds of cancers as an adjuvant in China [9]. A large number of literature have reported that matrine displayed cytotoxicity in a range of cancers, including lung cancer [10], breast cancer [11], liver cancer [12], and nasopharyngeal carcinoma [13]. Matrine has been recognized as an ideal lead compound for drug development due to its low toxicity, wide range of sources, and inexpensiveness. The main sites of structural modification on matrine are its C-14 and C-13, which could greatly improve its anticancer effects [14,15,16].

Recently, literature has reported that matrine could reverse drug resistance in breast cancer [17], lung cancer [18], and bladder cancer [19]. The anti-chemoresistance effects of matrine and its derivatives against nasopharyngeal carcinoma cells have been rarely reported. In our previous study, matrine derivative with thiophene group displayed reversal effect in NPC-CDDP resistance cells in vitro [20]. However, the reversal effects of the derivative in vivo were not prominent. Thus, in the present study, we designed and synthesized matrine derivatives with new thiophene side chains to explore potential drugs that would show significant reversal effects in vitro and in vivo.

## 2. Results

### 2.1. Chemistry

In the present study, the synthetic routine was outlined in Scheme 1. Matrine (**1**) was selected as a lead compound and a series of 14-methylene thiomatrine derivatives was synthesized for their potential application as anti-chemoresistance agents. First, thiomatrine (**2**) was easily obtained via thionation of matrine (**1**) with Lawesson’s reagent in moderate yields [15]. Then, thiomatrine (**2**) was transformed into the corresponding thiomatrine derivatives **3a**–**3h** using lithium diisopropylamide (LDA) and aromatic aldehyde in THF [21]. These products **3a**–**3h** were purified by silica gel column chromatography using ethyl acetate and dichloromethane as gradient eluents and their structures were characterized by ^1^H-NMR, ^13^C-NMR, and high-resolution mass spectrometer (HRMS) analyses (Appendix A). It is noticeable that there are two configurations of the 8 synthesized compounds, the (*E*)- and (*Z*)-14-methylene thiomatrine derivatives in theory. In our previous work, a thiomatrine derivative (compound **3k**) was synthesized using the same synthetic route of the present study [15]. The single-crystal structure of compound **3k** was analyzed and the results indicated that the configuration of **3k** was *E*-form. The results suggested that the configurations of **3a**–**3h** were *E*-form. Another issue needs to be addressed is that although compound 3a and 3b were firstly synthesized in our previous work [15], the anticancer and CDDP resistant reversal activities of compound **3a** and **3b** were not evaluated against NPC cells and CDDP resistant NPC cells. Thus, in the present study, compounds **3a** and **3b** were involved.

### 2.2. In Vitro Cytotoxicity

Prior to determining the combined inhibitory effects against NPC CDDP-resistance cells, the cytotoxicity effects of these 8 derivatives were evaluated against three NPC cell lines (CNE2, HONE1, and HK-1) and CDDP resistance cell line (CNE2/CDDP) by using MTT (3-(4,5-dimethyl-2-thiazolyl)-2,5-diphenyl-2-*H*-tetrazolium bromide) assays. The results indicated that all the 8 derivatives displayed better anti-proliferative effects (IC_50_ ranging from 35 to 700 µM) than the parent compound matrine (IC_50_ more than 1000 µM). Notably, among the 8 derivatives, compound **3f** showed the most prominent proliferation inhibition effects in NPC and NPC/CDDP resistant cells with IC_50_ ranging from 30 to 100 µM (Table 1). Moreover, a liver normal cell LO2 was involved to evaluate the selectivity of these 8 compounds by using MTT assays. The results showed that these 8 compounds did not exhibit selectivity (Table 1).

### 2.3. Combined Inhibitory Effects Evaluation of 8 Derivatives with CDDP

Compounds regularly show synergy effects with a relatively low dosage range (IC_10_ to IC_30_) [22]. In order to preliminarily verify whether 8 compounds could increase cell sensitivity to CDDP, CNE2/CDDP cells were co-treated with CDDP (5 µM) and each compound (IC_10_), respectively. The cytotoxicity was evaluated by using MTT assays. The results indicated that only compound **3f** could increase the inhibitory effects of CDDP against CNE2/CDDP cells (Figure 1). Thus, compound **3f** was selected for further experiments.

### 2.4. Synergistic Inhibitory Effect Evaluation of Compound **3f** with CDDP

To determine whether compound **3f** display synergistic inhibitory effects with CDDP against CNE2/CDDP cells, an MTT assay was performed. Cells were co-treated with compound **3f** and CDDP or treated with CDDP or **3f** alone. There are 9 groups of the combination between **3f** and CDDP (Table 2). Inhibition rates were calculated, and the fraction affected (Fa) and CI (Combination Index) were generated for each group, and dose–effect curves were obtained. The CI greater than, equal to, and less than 1.0 indicate antagonistic, additive and synergistic effects, respectively. The results showed that in CNE2/CDDP cells treated simultaneously with **3f** and CDDP, the CI values were less than 1, indicating synergism between **3f** and CDDP (Figure 2).

### 2.5. Combination Treatment of Compound **3f** with CDDP Potentiated Apoptosis in CNE2/CDDP Cells

Subsequently, to further verify whether the synergistic effects of compound **3f** with CDDP involved cell death, an apoptosis assay was conducted. CNE2/CDDP cells were co-treated with CDDP and compound **3f**, or CDDP and **3f** alone for 24 h. Then, cells were harvested for staining by using an apoptosis kit, following FACS detection (Figure 3a). Then, the data from three independent experiments were analyzed and the results were shown in the form of histogram (Figure 3b). The results indicated that combined treatment with CDDP (4 µg/mL) and **3f** (40 µM) increased significant cell apoptosis compared with CDDP (15.53 ± 1.61 vs. 7.58 ± 0.82) or **3f** (15.53 ± 1.61 vs. 9.97 ± 1.36) treatment alone (Figure 3b).

### 2.6. Molecular Docking of Compound **3f** with Anti-Apoptosis Protein Bcl-w

The pro-survival protein Bcl-w has been demonstrated to contribute to reduced cell apoptosis under cytotoxic conditions [23]. To better understand the mechanism of compound **3f**, molecular docking was carried out against Bcl-w (2Y6W). The results showed that the nitrogen atom of **3f** formed a hydrogen bond with glycine (gly90) around the active pocket, and the thiophene ring formed a π-cation interaction with glycine (gly89 and gly90) on the side chain. Besides, the chloride atom on the compound group forms hydrophobic interaction with glutamic acid (Glu), arginine (ARG), leucine (Leu), and proline (pro) around the pocket (Figure 4a,b).

### 2.7. In Vivo Anticancer Evaluation of Compound **3f**

Finally, to evaluate the combination inhibitory effects of compound **3f** with CDDP in vivo, the subcutaneous tumor-bearing model was applied by using strain of BALB/C nude mice. In detail, 32 BALB/C nude mice without tumor were purchased and CNE2/CDDP cells (2 × 10^7^) were subcutaneously injected into the right flank of each mouse to make tumor burden. Then, the 32 mice with tumor burden were divided into four groups which receive vehicle, CDDP (5 mg/kg), compound 3f (40 mg/kg), and CDDP with **3f** combination treatments (5mg/kg for CDDP and 40 mg/kg for **3f**), respectively, through intraperitoneal injection with two times a week for one month. The results implied that the combined treatment of CDDP with **3f** significantly reduced tumor volume and weight compared with CDDP or **3f** treatment alone (Figure 5a–c). Meanwhile, we noticed that there was no significant difference in body weight among mice in different groups (Figure 5d). The adverse effects of treatment were also tested by detecting serum concentration of Alanine Aminotransferase (ALT), Aspartate aminotransferase (AST), and creatinine (Cr). The results demonstrated that there were no significant changes in the serum concentrations of ALT, AST, and Cr in each group (Figure 5e). These results inferred that compound **3f** displayed favorable synergistic effects against CNE2/CDDP cells in vivo with no obvious side effects.

## 3. Discussion

It has been reported that matrine could reverse drug resistance in leukemia cells [24], breast cancer cells [17], and bladder cancer cells [19] in vitro with a high dosage of more than 1 mM. Accordingly, it can be inferred that the drug-resistant reversing effects of matrine with relative low dosage could be improved by structure modification. Thiophene is a heterocyclic scaffold containing sulfur which is generally present in many drugs on the market, such as diuretic, antiasthma, and anticancer drugs [25,26]. Some thiophene compounds have displayed favorable cytotoxicity activities [27,28]. Therefore, introducing thiophene structure into matrine would improve its drug reversing effects. In our previous study, a matrine derivative containing thiophene structure showed moderate drug resistance reversal effect against NPC-CDDP cells (HONE1/CDDP) in vitro but not in vivo [20]. In this study, 6 new matrine derivatives with thiophene side chain were inducted into the structure of matrine. Interestingly, we discovered that compound **3f** exhibited prominent synergistic inhibitory effects with CDDP against NPC-CDDP cells in vitro and in vivo (Figure 2 and Figure 5) with a relatively low dosage (20~60 µM) compared with that of matrine (>1 mM) in leukemia, breast, and bladder cancer cells [17,19,24]. It is also reported that thiophene compounds are limited in clinical use due to their risk of high toxicity [29]. Thus, we performed preliminary toxicity assessment experiments of **3f** in vivo by testing the level of ALT, AST, and Cr. The results revealed that there was no obvious toxicity in the physiological state with a dose of 40 mg/kg reflected by mouse body weight and ALT, AST, and Cr detection (Figure 5d–e). The present work is a very basic preclinical study that is far from clinical use. Whether compound **3f** could be used in human patients and what kind of dosage should be used needs to be further investigated.

Molecular docking technology can be used to study the interactions between drugs and targets, and predict their binding mode and affinity. Our docking results indicated that the chloride atom on the compound group can form hydrophobic interactions with glutamic acid (Glu), arginine (ARG), leucine (Leu), and proline (pro) around the active pocket (Figure 4a,b). Moreover, the docking assays of the other 7 derivatives and matrine with Bcl-w were conducted. The results indicated that all the 8 derivatives showed a stronger interaction than matrine and compound **3f** displayed the strongest interaction than the other 7 derivatives with a minimum London dG value (Appendix A), indicating that compound **3f** has the strongest affinity with Bcl-w which could inhibit the activity of Bcl-w. These results may explain why compound **3f** showed synergistic inhibitory effects with CDDP. However, to elucidate the interactions between compound **3f** and Bcl-w, further experiments are needed.

## 4. Experimental

### 4.1. Chemistry Synthesis

The melting points were measured on a WRS-1B micro melting point apparatus and were uncorrected. The ^1^H-and ^13^C-NMR spectra were obtained with Advance III HD 600 nuclear magnetic resonance (NMR) spectrometer (Bruker, Uster, Switzerland) operating at 600 MHz (for ^1^H) and 150 MHz (for ^13^C) in CDCl_3_ solution. Column chromatography was carried out using 300–400 mesh size silica gel (Qingdao Marine Chemical Factory, Qingdao, China). Reaction progress and target was monitored by TLC and visualized under UV light (254 nm). Matrine (95%) was purchased from Shanxi Undersun Biomedtech Co., Ltd. (Xi’an, China). All other reagents were of analytical grade and were purchased from Shanghai Energy Chemical Reagent Company (Shanghai, China) and were used without further purification.

First, Lawesson’s reagent (0.5 eq, 5 mmol) was mixed with toluene (100 mL). After adding 2.48 g (10 mmol) of matrine (**1**), the reaction mixture was stirred at 100 °C for 3 h. The excess solvent was evaporated off. The residue was purified by silica gel flash column chromatography (petroleum ether/ethyl acetate, 2:1) to afford thiomatrine (**2**) in 50% yields. Then, thiomatrine (2, 0.528 g, 2 mmol) was dissolved in anhydrous tetrahydrofuran (20 mL) in ice bath for 10–15 min under N_2_ protection. Lithium diisopropylamide (LDA, 5 mmol) was added dropwise into the mixture and the reactants were stirred at room temperature for 30 min. Aromatic aldehyde (5 mmol) was added in ice-salt bath, and then the mixture was refluxed at room temperature for 3 h. Water (10 mL) was added to quench this reaction. The residue was washed three times with dichloromethane (20 mL × 3). The organic phase was dried with anhydrous Na_2_SO_4_ and then the solvents were evaporated off. Purification of the crude products by silica gel column chromatography (ethyl acetate /dichloromethane 1:2) yielded 8 thiomatrine derivatives among which **3a** and **3b** have been reported previously [15] and **3c**~**3h** are reported for the first time. The purity of **3a**~**3h** has been tested by HPLC, which showed ~90% purity for **3a**~**3h**.

### 4.2. Characterization of **3c**~**3h**

The characteristic data of **3a** and **3b** were described previously [15].

*14-(thiophen-3-ylmethylene)thiomatrine (**3c**)* Light yellow powder, m.p. 166.8–167.5 °C, yield: 67%. ^1^H-NMR (600 MHz, Chloroform-*d*) δ 8.24 (s, 1H), 7.36–7.31 (m, 2H), 7.18 (d, *J* = 4.9 Hz, 1H), 5.60 (dd, *J* = 12.1, 4.0 Hz, 1H), 4.18 (dt, *J* = 10.9, 6.9 Hz, 1H), 3.67 (t, *J* = 12.3 Hz, 1H), 2.93 (ddd, *J* = 14.3, 7.6, 3.8 Hz, 1H), 2.85 (dd, *J* = 25.2, 11.3 Hz, 2H), 2.47 (ddd, *J* = 14.4, 10.4, 3.8 Hz, 1H), 2.18 (tdd, *J* = 10.5, 8.5, 6.9, 4.2 Hz, 2H), 2.03 (ddd, *J* = 15.4, 7.8, 4.0 Hz, 3H), 1.89–1.83 (m, 1H), 1.76 (ddd, *J* = 21.7, 17.8, 13.3 Hz, 3H), 1.62 (ddt, *J* = 12.7, 8.5, 4.4 Hz, 3H), 1.47 (ddd, *J* = 21.8, 10.3, 5.4 Hz, 3H). ^13^C-NMR (151 MHz, Chloroform-*d*) δ 193.47, 137.93, 135.11, 135.02, 128.94, 126.09, 125.39, 63.50, 57.11, 57.00, 53.25, 43.49, 35.97, 27.77, 26.28, 25.63, 22.35, 21.23, 20.80. HRMS (ESI) *m*/*z*: calcd for C_20_H_27_N_2_S_2_ [M + H]^+^: 359.1610; found 359.1620.

*14-((5-methylthiophen-2-yl)methylene)thiomatrine (**3d**)* Yellow powder, m.p. 186.2–187.4 °C, yield: 54%. ^1^H-NMR (600 MHz, Chloroform-*d*) δ 8.40 (s, 1H), 7.08 (s, 1H), 6.71 (s, 1H), 5.54 (d, *J* = 11.9 Hz, 1H), 4.16 (d, *J* = 7.2 Hz, 1H), 3.62 (t, *J* = 12.2 Hz, 1H), 2.83–2.76 (m, 3H), 2.48 (s, 3H), 2.40 (d, *J* = 18.2 Hz, 2H), 2.14 (s, 2H), 1.99 (d, *J* = 9.4 Hz, 5H), 1.72–1.67 (m, 3H), 1.45 (d, *J* = 12.8 Hz, 4H). ^13^C-NMR (151 MHz, Chloroform-*d*) δ 192.89, 143.74, 137.56, 134.84, 132.87, 131.26, 125.86, 63.55, 57.01, 56.97, 56.91, 53.56, 42.97, 36.13, 27.80, 26.33, 24.70, 22.30, 21.27, 20.82, 15.62. HRMS (ESI) *m*/*z*: calcd for C_21_H_29_N_2_S_2_ [M + H]^+^: 373.1767; found 373.1779.

*14-((3-methylthiophen-2-yl)methylene)thiomatrine (**3e**)* Yellow powder, m.p. 193.6–195.0 °C, yield: 71%. ^1^H-NMR (600 MHz, Chloroform-*d*) δ 8.57 (s, 1H), 7.30 (d, *J* = 4.6 Hz, 1H), 6.89 (d, *J* = 4.6 Hz, 1H), 5.55 (d, *J* = 11.9 Hz, 1H), 4.16 (d, *J* = 6.5 Hz, 1H), 3.63 (t, *J* = 12.4 Hz, 1H), 2.80 (dd, *J* = 23.5, 10.3 Hz, 3H), 2.36 (s, 3H), 2.20–2.13 (m, 4H), 1.98 (s, 5H), 1.73 (d, *J* = 12.9 Hz, 3H), 1.45 (d, *J* = 12.2 Hz, 4H). ^13^C-NMR (151 MHz, Chloroform-*d*) δ 193.25, 141.42, 132.99, 132.83, 132.13, 130.29, 126.66, 63.53, 56.99, 56.93, 56.88, 53.52, 43.09, 36.07, 27.77, 26.29, 25.02, 22.31, 21.24, 20.79, 14.93. HRMS (ESI) *m*/*z*: calcd for C_21_H_29_N_2_S_2_ [M + H]^+^: 373.1767; found 373.1778.

*14-((5-chlorothiophen-2-yl)methylene)thiomatrine (**3f**)* White powder, m.p. 226.4–227.6 °C, yield: 75%. ^1^H-NMR (600 MHz, Chloroform-*d*) δ 8.32 (d, *J* = 6.3 Hz, 1H), 7.04 (s, 1H), 6.89–6.85 (m, 1H), 5.51 (d, *J* = 3.8 Hz, 1H), 4.17 (s, 1H), 3.63 (dd, *J* = 14.9, 8.0 Hz, 1H), 2.81 (d, *J* = 7.6 Hz, 3H), 2.48 (d, *J* = 3.8 Hz, 1H), 2.14 (s, 2H), 2.01–1.96 (m, 3H), 1.83 (s, 1H), 1.70 (s, 3H), 1.56 (d, *J* = 14.3 Hz, 3H), 1.45 (d, *J* = 6.7 Hz, 3H). ^13^C-NMR (151 MHz, Chloroform-*d*) δ 192.34, 138.18, 133.67, 132.99, 132.76, 131.42, 126.58, 63.46, 56.98, 56.98, 56.95, 53.61, 43.04, 36.14, 27.78, 26.32, 24.63, 22.49, 21.24, 20.79. HRMS (ESI) *m*/*z*: calcd for C_20_H_26_N_2_ClS_2_ [M + H]^+^: 393.1220; found 393.1235.

*14-((4-bromothiophen-2-yl)methylene)thiomatrine (**3g**)* Yellow powder, m.p. 245.7–246.3 °C, yield: 57%. ^1^H-NMR (600 MHz, Chloroform-*d*) δ 8.35 (s, 1H), 7.31 (s, 1H), 7.16 (s, 1H), 5.56 (dd, *J* = 12.1, 4.0 Hz, 1H), 4.22 (dt, *J* = 11.8, 6.6 Hz, 1H), 3.68 (t, *J* = 12.3 Hz, 1H), 2.95–2.80 (m, 3H), 2.54 (ddd, *J* = 14.5, 9.7, 4.0 Hz, 1H), 2.26–2.17 (m, 2H), 2.06–1.98 (m, 3H), 1.88 (d, *J* = 14.1 Hz, 1H), 1.74 (dt, *J* = 16.3, 5.6 Hz, 3H), 1.65–1.57 (m, 3H), 1.48 (ddd, *J* = 22.0, 11.5, 3.6 Hz, 3H). ^13^C-NMR (151 MHz, Chloroform-*d*) δ 192.30, 140.58, 134.64, 132.93, 132.18, 124.97, 110.77, 63.45, 57.00, 56.98, 56.97, 53.57, 43.14, 36.10, 27.77, 26.31, 24.75, 22.39, 21.23, 20.78. HRMS (ESI) *m*/*z*: calcd for C_20_H_26_N_2_BrS_2_ [M + H]^+^: 439.0695; found 439.0710.

*14-((5-bromothiophen-2-yl)methylene)thiomatrine (**3h**)* Yellow powder, m.p. 239.9–240.6 °C, yield: 64%. ^1^H-NMR (600 MHz, Chloroform-*d*) δ 8.40 (d, *J* = 1.7 Hz, 1H), 7.06 (t, *J* = 3.5 Hz, 2H), 5.57 (dd, *J* = 12.1, 4.0 Hz, 1H), 4.23 (dt, *J* = 10.8, 6.4 Hz, 1H), 3.68 (t, *J* = 12.3 Hz, 1H), 2.90–2.81 (m, 3H), 2.56–2.49 (m, 1H), 2.26–2.17 (m, 2H), 2.06–2.00 (m, 3H), 1.88 (dt, *J* = 13.9, 2.4 Hz, 1H), 1.76 (tdd, *J* = 16.7, 9.3, 5.0 Hz, 3H), 1.66–1.57 (m, 3H), 1.52–1.45 (m, 3H). ^13^C-NMR (151 MHz, Chloroform-*d*) δ 192.40, 141.03, 133.54, 132.96, 132.25, 130.25, 115.74, 63.49, 57.00, 56.99, 56.97, 53.63, 43.04, 36.13, 27.77, 26.32, 24.62, 22.54, 21.24, 20.79. HRMS (ESI) *m*/*z*: calcd for C_20_H_26_N_2_BrS_2_ [M + H]^+^: 439.0695; found 439.0713.

### 4.3. Cell Viability Assay

Nasopharyngeal carcinoma cells (CNE2, HONE1, and CNE2/CDDP) were purchased from Shanghai Aulu Biological Technologh Co., Ltd. (Shanghai, China). HK-1 cell was purchased from Shanghai Shunran Biological Technologh Co., Ltd. CNE2, HONE1, and HK-1 were cultured in DMEM medium supplemented with 10% fetal bovine serum, 1% penicillin-streptomycin. CDDP-resistant NPC cell CNE2/CDDP was cultured in DMEM medium supplemented with CDDP to sustain its drug resistance. Cell viability was detected using the MTT assay. All the 8 derivatives were resolved with DMSO to make a storage solution at 1 M, and CDDP was resolved with phosphate buffer saline (PBS) to make a storage solution at 100 mM. The maximum concentrations of these 8 derivatives and CDDP used in MTT assays were set as 1 mM and 100 µM diluted with DMEM medium, respectively. Medium with 0.1% DMSO was set as control. To begin, an initial MTT assay was conducted to evaluate the cytotoxicity of **3a**~**3h** with a dose range of 50, 100, 200, 400, and 800 µM. Then, a more precise dose range was set to calculate the IC_50_ of each compound (For **3a**~**3e**: 50, 100, 200, 400, 600, and 800 µM; for **3f**~**3h**: 12.5, 25, 50, 100, 200, and 400 µM). Then, 5000 cells were seeded in a 96-well plate for each well and treated with respective compounds at different concentrations for 72 h following additional treatment with MTT for 2~3 h. Then, MTT was removed and 150 µL DMSO was added into each well and the absorption values were measured at 490 nm using a SpectraMAX M5plate reader (Silicon Valley, CA, USA).

### 4.4. Synergistic Effects Calculation

The combination index (CI) was evaluated using Calcusyn 2.0 based on the Chou-Talalay method. Data from cell viability assays were applied to the CI calculation. CI < 1 suggests synergism, CI = 1 suggests an additive effect, while CI > 1 suggests antagonism.

### 4.5. Apoptosis Assay

CNE2/CDDP cells (3 × 10^5^) were seeded into a 6-well plate for each well. Twenty-four hours later, medium was removed and cells were treated with different compounds for 72 h. Then, cells were harvested, washed with cold PBS, and stained with Annexin V-FITC and PI for 15 min in dark. Finally, samples were analyzed by flow cytometer. 

### 4.6. Molecular Docking

The molecular docking between compound **3f** and Bcl-w (PDB ID: 2Y6W) was carried out via applying the Molecular Operating Environment (MOE) software version 2008 as described previously [30]. Firstly, the molecular structure of compound **3f** was plotted by using ChemDraw. Then, the structure of compound **3f** was subjected to energy minimization using HyperChem via molecular dynamics (mm +) analysis, and the force field partial charges were calculated for each molecule. After 3D protonation and energy minimization, the protein active sites of Bcl-w were defined by molecular pore technology for molecular docking. Stochastic conformational analysis was run for compound **3f** using default settings and the most stable conformer were retained.

### 4.7. In Vivo Mouse Model

The in vivo studies were conducted according to protocols approved by the Animal Ethics Committee of Guangxi University (GXU2018-019). The subcutaneous tumor-bearing model using BALB/C nude mice was applied to assess the in vivo CDDP-resistant reversal activity of compound 3f. In brief, 32 BALB/C nude mice without tumor were purchased and CNE2/CDDP cells (2 × 10^7^) were subcutaneously injected into the right flank of each mouse to make tumor burden. Then, the 32 mice with tumor burden were divided into four groups which receive vehicle, CDDP (5 mg/kg), compound 3f (40 mg/kg), and CDDP with **3f** combination treatments (5 mg/kg for CDDP and 40 mg/kg for **3f**) (*n* = 8 for each group; twice a week for a month). Drugs were resolved with sterile saline–ethanol–DMSO (89:10:1, *v*/*v*/*v*). Sterile saline–ethanol–DMSO (89:10:1, *v*/*v*/*v*) was used as vehicle. Tumor volume was calculated according to the formula: V = 0.5 × L × W^2^ (L: the long diameter of the tumor; W: the short diameter of the tumor). At the time of the animal sacrifice, tumors were excised and weighted.

### 4.8. ELISA Assays

Mice were sacked at the time of the animal sacrifice and blood was collected to detect ALT, AST, and Cr. Immediately, serum was separated by centrifugation at 2000 g, and ALT, Cr, and AST were determined using ELISA kits (Shanghai Enzyme-linked Biotechnology Co., Ltd., catalog: ml063179, ml058577, and ml057581) according to the manufacture instructions. The absorbance of the plates was read at 450 nm using an automated microplate reader (Bio-Tek, Winooski, VT, USA).

### 4.9. Statistical Analysis

In the present study, data are present as the mean ± SD. Experiments were conducted for three independent times. *P* values < 0.05 were considered as statistically significant by using Student’s *t*-test of unpaired data.

## 5. Conclusions

Eight matrine derivatives containing thiophene group were designed and synthesized. The cytotoxicity effects of these 8 compounds against nasopharyngeal carcinoma (NPC) cells and cisplatin-resistance NPC cells (CNE2/CDDP) were detected and compound **3f** could induce apoptosis and display potent synergistic inhibitory effects with cisplatin against CNE2/CDDP cell in vitro and in vivo. 

## Data Availability

The data presented in this study are available on request from the corresponding author.

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
