# Peer review of "Design, Synthesis, Molecular Docking, and Tumor Resistance Reversal Activity Evaluation of Matrine Derivative with Thiophene Structure"

_molecules, 2021, doi:10.3390/molecules26020417_

Round 1

Reviewer 1 Report

41- where is marine approved?

43-44- I'm not sure I'd say matrine has druggable advantages. There are obvious synthesis limitations that they even note, ie the structure is not readily altered and therefore optimization is difficult.  There's also a major issue with auto-oxidation to oxymatrine which was not discussed.

56- The synthesis looked fine, but there were no comments about verifying the olefin stereochemistry.

86- The synergism studies are interesting, but not very well presented. Is there a reason the data was not presented as in Table 1?  The entire section could be explained much better.  Several acronyms aren't defined (MTT, CI, etc). Table 2 is complicated and redundant. Fig 2 is not well presented.

106- I understand the bar graph of Fig 3 but what is the other part of the figure?  It's not explained in the text or under the Figure.

109- The entire molecular docking section is a bit questionable. Why did they pick BCl?  Does matrine bind there?  Does cis-platin?  It's hard to tell, but the 3D structure of the analog does not seem to be correct.  Matrine exists as a U-shaped structure with the 2 nitrogens almost on top of each other (which is what facilitates auto-oxidation. The structure seems a bit flat for that.

120- The in vivo studies look interesting, but are not well explained.  Do these mice have tumors?  The authors just say nude mice, but there are several kinds of mice.

140- The discussion in general has the most issues with English and needs to be carefully edited.

150- It's probably good to state that 2 compounds were already known earlier in the paper because the first time this is implied is in the discussion, and even then I didn't realize that until the synthesis section.

159- The last part of the discussion needs to make sense with whatever the authors decide to do with the modeling. It may make sense to dock with Bcl but that needs to be better explained.

4.1- The synthesis section has several typos including diisopropylamine and ice-salt bath.

4.6- A little more information about the docking method would be good.

4.7- There was little information about the specific mouse strain and tumor model.

4.8-  I would not use "executed to death" in that description.

Reviewer 2 Report

In this manuscript, the authors reported the first-step essential characterization of their newly synthesized compounds. The work is well done and clearly presented.

I only have few minor comments.

It is not immediately clear from the text which assay they used to determine the IC50s (Table1). I suggest indicating in the text that it was MTT. Exclusively pointing out that they used MTT for experiments of table2/figure2 might create confusion.

Legend to Table 1 misses the word “resistant” in the description of CNE2/CDDP cell line.

Figure 2: Why did the authors use only the IC10 for combination experiments?

Figure 5: How do they determine in vivo concentration of CDDP and 3f compound?

Based on previous studies made with matrine, is the 3f dosage used in mice achievable in humans? Authors should also comment on the possible way of administration of 3f in patients.

Reviewer 3 Report

The paper is rather typical "make new derivatives check the activity and publish" and put next one on the shelv.

The procedures are typical quite well documented but originality is not great.

Synthesis also is not "sophisticated".Not sufficient explanation convicing why performed modification makes the componds more active.

Reviewer 4 Report

The paper entitled " Design, synthesis, molecular docking and tumor resistance reversal activity evaluation of matrine derivative with thiophene structure” contributed by Lichuan Wu and co-authors reports the synthesis of 8 thiophene thiomatrine derivatives (obtained by a methodology previously described by the team) and their activity against nasopharyngeal carcinoma (NPC) cells and cisplatin-resistant NPC cells.
The study is well organized but presents some important drawbacks that must be addressed. The most important are the low purity of some of the 8 compounds observable on NMR spectra presented, and the absence of test of the synthetized compounds against a normal cell line to address the compounds selectivity.

In detail:

Line 40 – Reference 9 does not refer matrine approval to treat non-small cell lung cancer and liver cancer as an adjuvant. Other papers refer the approval but it is not a worldwide approval but local (China) approval. The statement should be clarified. The paper “Matrine: A Promising Natural Product With Various Pharmacological Activities - https://doi.org/10.3389/fphar.2020.00588” has a clear presentation of the matrine uses and should be included in the references list.

Line 59 – Ref 23 cited here is a paper were thioamides were prepared from amides using Lawesson´s reagent. Thiomatrine has been prepared from matrine and Lawesson's reagent in 2018 in ref 16. This reference 16 should be used instead of 23.

Line 63 and supplementary material - It is possible to observe in the spectra of the compounds 3a-h that they are far from pure (by example it can be easily seen in 13CNMR of 3e a large amount of a second compound). How the authors can assure that the observed biological activity does not result from the impurities? Are the impurities characterized and their biological activity evaluated? This issue should be addressed.

Table 1 – A comparison with a normal cell line should be included to evaluate the selectivity of the compounds before the in vivo tests.

Line 114 - I cannot understand here are the chlorine ions present. How the chlorine atom connected to the thiophene ring in compound 3f improve his affinity to the pro-survival protein?

Bibliography – It should be carefully revised since there are journal names abbreviated (ex: ref 14) and others without abbreviations (ex. ref 11).

Round 2

Reviewer 1 Report

So if I understand the response, the authors synthesized mixtures of olefins (3a-3h)?  That is a bit different than what was discussed originally and if that's the case, then the ratios need to be described since they are different compounds and both tested. It should have been relatively easy to separate, so makes me less enthusiastic.

All of the chemistry issues (placed in the back) need to be discussed on the chemistry section.  I'm not sure why the authors decided to put it into the discussion.  Compounds 3a-3h were synthesized as a mixture of olefins.

If mixtures are being tested, I'm not sure it is appropriate to describe things as "compound 3f".  It's really a mixture of two compounds.  That needs to be addressed throughout the manuscript.

There is no longer a chart describing what 1 through 9 mean.  The synergism section (2.4) needs to be improved.

Line 99:  It should read "an MTT assay was performed" I believe.  There are English issues throughout the manuscript and it needs to be heavily edited (which I can not do as a PDF).

Figure 3.  So the left is the data, and the right is a graphic with the data obtained from the data on the left?  That needs to be described better, it's not 'statistics' it's quantified results.

Section 2.7:  The authors should say the injection creates a tumor to make it more explicit, similar to what they wrote in the response to reviewers.

It's probably best to say "mice were sacked" on line 310.

Author Response

1). So if I understand the response, the authors synthesized mixtures of olefins (3a-3h)?  That is a bit different than what was discussed originally and if that's the case, then the ratios need to be described since they are different compounds and both tested. It should have been relatively easy to separate, so makes me less enthusiastic.

All of the chemistry issues (placed in the back) need to be discussed on the chemistry section.  I'm not sure why the authors decided to put it into the discussion.  Compounds 3a-3h were synthesized as a mixture of olefins.

If mixtures are being tested, I'm not sure it is appropriate to describe things as "compound 3f".  It's really a mixture of two compounds.  That needs to be addressed throughout the manuscript.

-- We deeply appreciate the reviewer’s comments for compound configuration. In our previous work, a thiomatrine derivative (compound 3k) was synthesized using the same synthetic route of the present study (Ref 15. Li, Z.; Wu, L. C.; Cai, B.; Luo, M. Y.; Huang, M. T.; Rashid, H. U.; Yang, Y. W.; Jiang, J.; Wang, L. S., Design, synthesis, and biological evaluation of thiomatrine derivatives as potential anticancer agents. Med Chem Res 2018, 27, (8), 1941-1955). The single crystal structure of compound 3k was analyzed and the results indicated that the configuration of 3k was E-form (Figure 2 and table 1 of ref 15). Base on the single crystal data of compound 3k from our previous work (ref 15) which was synthesized using the same synthetic route, we speculate that the configurations of 3a~3h in the present study were E-form. These results were described on page 2, line 64-69 of revised manuscript. And as suggested, the description of compound configuration in the discussion part has been removed.
2).There is no longer a chart describing what 1 through 9 mean.  The synergism section (2.4) needs to be improved.

-- Many thanks. As suggested, a new table named table 2 was created on page 5 to better explain the synergism section. In the table 2, 1 through 9 were described in details. Also, the section of 2.4 was re-written on page 4, line 105-112 of the revised manuscript.

3). Line 99:  It should read "an MTT assay was performed" I believe.  There are English issues throughout the manuscript and it needs to be heavily edited (which I can not do as a PDF)..

-- Thank you so much for pointing out the mistakes. As suggested, more appropriate descriptions have been made throughout the manuscript, such as on page 4, line 105, page 6, line 139 and page 8, line 196 of revised manuscript.

4) Figure 3.  So the left is the data, and the right is a graphic with the data obtained from the data on the left?  That needs to be described better, it's not 'statistics' it's quantified results.

-- Many thanks. The figure 3 has been described in detail on page 5, line 122-126.

5) Section 2.7:  The authors should say the injection creates a tumor to make it more explicit, similar to what they wrote in the response to reviewers.

-- Thank you so much. The section 2.7 was described similar to what we described in the response to reviewers on page 6 and 7, line 146-152 of revised manuscript.

6) It's probably best to say "mice were sacked" on line 310.

-- Many thanks. The description has been corrected on page 11, line 336.

Reviewer 3 Report

still authors did not responded clearly to my previous:

"Not sufficient explanation convicing why performed modification makes the componds more active.

Some structure activity should be performed. Why compound 3f is active and others are not. The docking was performed only for 3f and explanation for it is provided. Why the  docking for rest 7 compounds are nor done. 

At least the docking for less active compounds should be done showing that their iinteraction is worse and correlate with the change of activity

Author Response

Still authors did not respond clearly to my previous: "Not sufficient explanation convincing why performed modification makes the componds more active.

Some structure activity should be performed. Why compound 3f is active and others are not. The docking was performed only for 3f and explanation for it is provided. Why the docking for rest 7 compounds are not done.

At least the docking for less active compounds should be done showing that their iinteraction is worse and correlate with the change of activity

-- We really appreciate the reviewer’s comments. As suggested, the docking of the other derivatives and matrine with Bcl-w were conducted. The results indicated that that all the 8 derivatives showed a stronger interaction than matrine and compound 3f displayed the strongest interaction than the other 7 derivatives with a minimum London dG value (Supplementary Table 1), indicating that compound 3f has the most strongest affinity with Bcl-w which could inhibit the activity of Bcl-w. These results may explain why compound 3f showed synergistic inhibitory effects with CDDP. However, to elucidate the interactions between compound 3f and Bcl-w, further experiments are needed. These description was made on the discussion part on page 8, line 195-205.

Supplementary Table 1

Number Compound London dG
1 3f -10.9041214
6 3a -10.629405
4 3d -10.5346899
7 3g -10.5202732
3 3c -10.088953
8 3h -10.0790882
2 3b -9.92576027
5 3e -9.87210655
9 matrine -9.71206188

Reviewer 4 Report

I still have concerns relatively to the impurities of compounds 3a-h. Using as example compound 3e for which one the authors on they answer present an HPLC showing the main compound with a 88% peak area and attributing the second peak to thiomatrine. It sems unlikely since thiomatrine has no CC double bonds and the presented 13C NMR presented has several signals between 120 and 140 pmm besides de 6 signal attributed to 3e compound. If thiomatrine a third compound must be present and it can be proposed to be a result of the pair of E/Z diastereomers. All this clarifications should be included in the paper on the chemistry sub chapter of Results an in the Discussion here the question E/Z diastereomers was introduced (line 173).

In line 128 authors refer “Besides, the chloride ions on the compound group form hydrophobic interaction with glutamic acid (Glu), arginine (ARG), leucine (Leu) and proline (pro) around the pocket (Figure 4a and 4b).”. – “chloride ions” should be replaced by “chloride atoms” since the compound 3f is not a salt.

Author Response

Still authors did not respond clearly to my previous: "Not sufficient explanation convincing why performed modification makes the componds more active.

Some structure activity should be performed. Why compound 3f is active and others are not. The docking was performed only for 3f and explanation for it is provided. Why the docking for rest 7 compounds are not done.

At least the docking for less active compounds should be done showing that their iinteraction is worse and correlate with the change of activity

-- We really appreciate the reviewer’s comments. As suggested, the docking of the other derivatives and matrine with Bcl-w were conducted. The results indicated that that all the 8 derivatives showed a stronger interaction than matrine and compound 3f displayed the strongest interaction than the other 7 derivatives with a minimum London dG value (Supplementary Table 1), indicating that compound 3f has the most strongest affinity with Bcl-w which could inhibit the activity of Bcl-w. These results may explain why compound 3f showed synergistic inhibitory effects with CDDP. However, to elucidate the interactions between compound 3f and Bcl-w, further experiments are needed. These description was made on the discussion part on page 8, line 195-205.

Supplementary table 1

Number Compound London dG
1 3f -10.9041214
6 3a -10.629405
4 3d -10.5346899
7 3g -10.5202732
3 3c -10.088953
8 3h -10.0790882
2 3b -9.92576027
5 3e -9.87210655
9 matrine -9.71206188